# A Face-To-Face Comparison of Tumor Chicken Chorioallantoic Membrane (TCAM) In Ovo with Murine Models for Early Evaluation of Cancer Therapy and Early Drug Toxicity

**DOI:** 10.3390/cancers14143548

**Published:** 2022-07-21

**Authors:** Tristan Rupp, Christophe Legrand, Marion Hunault, Laurie Genest, David Babin, Guillaume Froget, Vincent Castagné

**Affiliations:** Porsolt SAS, Z.A. de Glatigné, 53940 Le Genest-Saint-Isle, France; clegrand@porsolt.com (C.L.); mhunault12@gmail.com (M.H.); lgenest@porsolt.com (L.G.); dbabin@porsolt.com (D.B.); gfroget@porsolt.com (G.F.); vincent.castagne53@sfr.fr (V.C.)

**Keywords:** chicken embryos, 3Rs guidelines, preclinical models, oncology, tumor CAM

## Abstract

**Simple Summary:**

Cancer remains a major health issue, and the development of new drug strategy is still mandatory. Currently, rodent models remain widely used in order to identify the first proof of concept of drug efficacy. Nevertheless, ethical considerations, cost, and time constraints highlight the need to develop alternatives to limit the use of conscious animals. Here, we showed in a face-to-face comparison that tumor-bearing eggs and tumor-bearing mice had a similar chemotherapy response in four different cancer models. We think that the so-called tumor chicken chorioallantoic membrane (TCAM) model may represent a relevant avenue for evaluating cancer treatment according to the 3Rs strategy in “The Principles of Humane Experimental Technique” aiming to reduce, refine, and replace animals to limit their number and suffering in experiments.

**Abstract:**

Ethical considerations, cost, and time constraints have highlighted the need to develop alternatives to rodent in vivo models for evaluating drug candidates for cancer. The tumor chicken chorioallantoic membrane (TCAM) model provides an affordable and fast assay that permits direct visualization of tumor progression. Tumors from multiple species including rodents and human cell lines can be engrafted. In this study, we engrafted several tumor models onto the CAM and demonstrated that the TCAM model is an alternative to mouse models for preliminary cancer drug efficacy testing and toxicity analysis. Tumor cells were deposited onto CAM, and then grown for up to an additional 10 days before chronic treatments were administered. The drug response of anticancer therapies was screened in 12 tumor cell lines including glioblastoma, melanoma, breast, prostate, colorectal, liver, and lung cancer. Tumor-bearing eggs and tumor-bearing mice had a similar chemotherapy response (cisplatin and temozolomide) in four human and mouse tumor models. We also demonstrated that lethality observed in chicken embryos following chemotherapies such as cisplatin and cyclophosphamide were associated with corresponding side-effects in mice with body weight loss. According to our work, TCAM represents a relevant alternative model to mice in early preclinical oncology screening, providing insights for both the efficacy and the toxicity of anticancer drugs.

## 1. Introduction

The modeling of human cancers for research in oncology is a central issue in the context of preclinical tests accompanying the development of new anticancer therapies. The major criteria considered in the development of animal models are the reliability of the model, the ethical concern, the speed of execution, and the cost of realization [1]. Currently, the typical animal models developed for studies in oncology are in mice [2]. The development of these models involves a relatively long production time and an associated high cost [3]. In addition, some types of cancer cells, such as human cancer cells, cannot be implanted in fully immunocompetent mice and require the use of immunodeficient mice lacking a functional immune system [2]. Motivated by ethical concerns, Russell and Burch developed the 3Rs strategy in “The Principles of Humane Experimental Technique” in order to reduce, refine, and replace animals and to limit the number and suffering of animals in experiments [4]. Today, there is still a need to use alternative models in order to improve cancer research.

One of the alternative models is the use of chicken embryos for cancer research. The eggs are used as a bioreactor with a dynamic microenvironment favorable to tumor development [5,6]. During avian embryo development, the mesoderm rapidly expands, generating the richly vascularized chorioallantoic membrane (CAM) and transforming the structure of the eggs. The CAM serves as respiratory organ for the chicken embryos. The CAM grows and merges with the chorion from embryonic day 3 (ED3). At this stage, the blood vessels into the CAM expand and come in close contact with the porous shell, allowing gas exchange. The vascularized membrane continues to grow until ED13 and covers the surface of the shell, permitting oxygen, nutrient, and mineral transport to the embryo. The lymphatic vessel network also develops in the CAM until ED9, but the immune system is not active before ED18 [7]. The chicken embryos are, therefore, not fully immunocompetent before this stage. Importantly, chicken embryos are not described as having functional nociception up to the later stages of development [8]. Finally, chicken embryogenesis lasts 21 days until hatching [7].

Tissue transplantation onto the CAM is feasible until ED18, due to the absence of fully immunocompetent system, representing an interesting model for carrying out in vivo experiments. The CAM assay has also been extensively characterized for studying angiogenesis and vascular remodeling [9,10,11]. It represents a versatile and relevant preclinical in vivo oncology model using tumor xenografts from a wide variety of cancers from lymphoma to solid tumors such as head and neck or hepatoma models [12,13,14,15]. The study of cell proliferation and migration can also be monitored in chicken embryos [16,17,18]. Importantly, the tumor CAM (TCAM) assay can be considered to be ethically more tolerable than rodent models, as the CAM is not innervated [19], thereby complying better with the 3Rs guidelines for animal usage in research. The TCAM assay represents a quick and relative low-cost model for efficacy and safety testing as well as screening of a large library of pharmacological substances [20]. Indeed, Eckrich and collaborators calculated that the cost of the TCAM assay is around 20-fold less than mouse models [3].

To our knowledge, no direct comparison between mice and chicken embryos has been published to monitor tumor response or safety issues following anticancer drug treatment. Here, we demonstrated that the TCAM assay can be a suitable alternative to rodents for the early evaluation of anticancer drugs. Our procedure used cells in suspension seeded within a mixture of Matrigel™ and medium to avoid cellular dispersal and provide nutrient support until the cells recruit a vascular supply. The tumor cells were deposited onto the CAM of the chicken embryo and generated tumors that were monitored over a 10 day period. A tumor typically takes advantage of the developed vascular network in order to grow. This model was used to evaluate five anticancer agents in dedicated tumor models. A total of 12 tumor cell lines originating from breast, colon, lung, liver, and brain tumors demonstrated various drug responses. We also demonstrated a positive relationship of drug response with two chemotherapeutic agents, cisplatin and temozolomide, between mouse and chicken embryo xenografts. Lastly, we compared the safety profile of two chemotherapeutic agents, cisplatin and cyclophosphamide, and demonstrated comparable responses in both mice and chicken embryos.

## 2. Material and Methods

### 2.1. Animals

Fertilized Leghorn white eggs were supplied by EARL des Bruyères (Dangers, France). Upon receipt, eggs (60 ± 10 g) were incubated horizontally at 37.7 °C and 60–70% humidity under rotation (embryonic development ED0) in egg incubators (Cikuma™). Housing was adapted according to [21]. The 6 week old female BALB/cAnN-Foxn1nu/nu/Rj (BALB/c-nude) mice or BALB/cJRj mice, supplied by Janvier Labs, were acclimated at least 5 days before the experiments. The implantation of tumor cells was performed on 7 or 8 week old mice. BALB/cJRj mice were housed up to 10 animals per cage in a biosafety level 1 laboratory. BALB/c-nude mice were housed in a biosafety level 2 laboratory and grouped to six animals per individually ventilated cage (NEXGEN MOUSE IVC™, Allentown^®^, PA, USA) on NestPak^®^ (Allentown^®^). Nesting enrichment was provided (tube, cotton, and wood). The laboratories were maintained under artificial lighting (12 h) between 7:00 a.m. and 7:00 p.m. in a controlled ambient temperature of 22 ± 2 °C and relative humidity of 30–70%. The number of animals per group included in each experiment is described in the legends of the corresponding figures.

### 2.2. Animal Ethical Consideration and Limit Points

The procedure was adapted from [21,22]. All methods, designed to minimize animal suffering and to ensure good quality of biological samples, were adapted from basic procedures commonly used in studies performed in rodents. Experiments were conducted in strict accordance with Council Directive No. 2010/63/UE of 22 September 2010 on the protection of animals used for scientific purposes, the French Decree No. 2013-118 of 1 February 2013 on the protection of animals for use and care of laboratory animals, and the recommendations of the Association for Assessment and Accreditation of Laboratory Animal Care (AAALAC agreement 001463 obtained in 2012 and renewed in June 2021). All experiments were also approved by the Internal Animal Care and Use Committee (IACUC) of Porsolt for animal experimentation (Porsolt’s agreement n°F53 1031) and followed the ARRIVE guidelines.

The tumor volume and body weight of the mice were measured and recorded two to three times per week. Tumor volume exceeding 2000 mm^3^, a weight loss greater than 20% relative to the initial weight of the animal, and tumor necrosis, including bleeding, ulceration, hypothermia (<34 °C), dyspnea, failure to eat and drink, loss of balance, or marked sedation, were considered as critical limitation points. When one of these conditions was met, mice were sacrificed by CO_2_ inhalation.

### 2.3. Cells and Cell Culture

CT26.WT (CT26) mouse colon carcinoma cells (CRL-2638™ obtained from ATCC^®^), HCT-8 human ileocecal colorectal adenocarcinoma cells (CCL-244™ obtained from ATCC^®^), HCT-116 human colorectal carcinoma cells (91091005 obtained from ECACC^®^), 4T1 triple0negative mouse breast carcinoma cells (CRL-2539™ obtained from ATCC^®^), MDA-MB-231 triple-negative human breast adenocarcinoma cells (HTB-26™ obtained from ATCC^®^), U118MG human glioblastoma cells (HTB-15™ obtained from ATCC^®^), GL261 mouse glioma cells (ACC 802 obtained from Leibniz Institute DMSZ), HepG2 human hepatocellular carcinoma cells (HB-8065™ obtained from ATCC^®^), PC-9 human lung adenocarcinoma cells (90071810 obtained from ECACC), PC-9 cisplatin-resistant (PC9/CR) human lung adenocarcinoma cells (obtained from Institut Bergonié, Bordeaux, France), LNCaP human prostate carcinoma cells (CRL-1740™ obtained from ATCC^®^), PC-3 human prostate grade IV adenocarcinoma cells (CRL-1435™ obtained from ATCC^®^), and A375 human melanoma cells (CRL-1619™ obtained from ATCC^®^) were cultured in vitro with RPMI 1640 (Gibco^®^, ATCC-formulated) supplemented with fetal bovine serum (FBS, Gibco^®^) at a final concentration of 10% and antibiotics (penicillin 100 U/mL–streptomycin 100 µg/mL, Gibco^®^). Cell lines were used for up to six passages from their original passage. All cell lines tested negative for mycoplasma just prior to the experimental sessions using MycoAlert^®^ Mycoplasma Detection Kit. Only mycoplasma-negative cell lines were used for the experimentation included in this study. The procedure was adapted from [21,22]. Cells were grown in cell incubator at 37 °C and 5% CO_2_. Before cell injection, 70–90% confluent cells were split, and cell viability was assessed using an automated cell counter, Nucleocounter NC-200™ (Chemotec^®^). The cell suspension was prepared according to the viable cell count. All procedures were performed in aseptic conditions, under a laminar flow hood.

### 2.4. Tumor CAM Assays

A total of 2 × 10^6^ cells (the quantity of cells was validated in prior pilot study; data not shown) for all tumor cell lines were deposited onto the CAM. Tumor cell inoculation was performed on day 7 of embryonic development (ED7) after opening and exposing the CAM. Rotation within the incubator was stopped at this time. Seven day old eggs were inspected using an egg candler to visualize and mark the vasculature of the CAM. Nonviable eggs, as indicated by nonperfused vessels, were removed. Using a sterile push pin, one hole (around 3 mm deep) was made at the narrow apex (in the air sac) and another one was made on the upper part of the eggs (around 1 mm deep, to avoid contact with the CAM). The CAM was then dropped away from the shell by applying suction against the air sac hole using a safety bulb. After successful dropping, a 2 cm^2^ opening (2 cm × 1 cm) was made using a drilling device. No albumen was extracted from the eggs. The shell piece was then carefully removed using sterile forceps and the window was sealed using masking tape. Eggs were placed back in the incubator until the time of inoculation. Meanwhile, cells were prepared and resuspended in ice-cold phosphate-buffered saline (PBS)/Matrigel (Matrix Basement Membrane Growth Factor-reduced and Phenol Red-free, Corning B.V.^®^, reference 356231) with a ratio of 1:1. For the inoculation, a silicon ring, detached from the cap of a sterile cryogenic vial, was dropped on the CAM, and 20 µL of cell suspension was pipetted into the center of the ring. The eggs were retaped and moved with caution back to the incubator for an additional 10 days (Figure 1A). At ED10, viable embryos, identifiable by CAM blood perfusion, were randomized, and treatment were started and applied topically directly onto the silicone ring-containing tumors (100 µL). At ED17, vascularized tumors could be observed. Before tumor dissection and analysis, chicken embryos were anesthetized for at least 10 min with a solution of ketamine (100 mg/kg) and xylazine (10 mg/kg) prior to sacrifice by incubating for 60 min with 1 mL of formalin 10% at room temperature under a hood. Finally, tumors were dissected, collected, and imaged. Drug efficacy was evaluated by the effect on tumor size and weight, as exemplified with MDA-MB-231 breast cancer cells treated with cisplatin that showed reduced tumor size (Figure 1B). Drug safety was evaluated by analysis of embryo survival.

### 2.5. Subcutaneous Graft Murine Models

A total of 5 × 10^5^ CT26 cells or 5 × 10^5^ 4T1 cells (the quantity of cells was validated in a prior pilot study; data not shown) were injected subcutaneously into the right flank of BALB/cJRj mice. A total of 5 × 10^6^ PC-9/CR or 5 × 10^5^ U118MG cells were injected into BALB/c-nude mice. The procedure was adapted from [21,22]. The cells to be implanted were resuspended in sterile PBS and kept on ice. Mice were placed under anesthesia with 2% isoflurane (Axience^®^, reference 152678) at 2 L/min on a warming pad and with eye lubricant during the procedure. The back of the mice was shaved, and the area for injection was cleaned with chlorhexidine (Antisept™, reference ANT015) before the injection of 100 µL of cell suspension using an insulin syringe. Mice were identified by permanent tattoo. Finally, the mice were monitored (breathing) until they woke up.

Tumor volume was measured two to three times per week with a caliper. The tumor volume was calculated using the formula V = (a^2^ × b)/2, where b is the longest axis and a is the perpendicular axis to b. Survival and body weight were also monitored throughout the study.

### 2.6. Treatments

Cisplatin was purchased from Santa Cruz^®^ (reference sc-200896, diluted in saline). Temozolomide was purchased from Selleckchem^®^ (reference S1237, diluted in 5% dimethyl sulfoxide (DMSO) + 30% Polyethylene glycol 300 (PEG 300) diluted in H_2_O). Doxorubicin hydrochloride was purchased from Sigma-Aldrich^®^ (reference D2975000, diluted in saline). 5-Fluorouracil (5-FU) was purchased from Sigma-Aldrich^®^ (reference F6627, diluted in saline). Sorafenib was purchased from Carbosynth^®^ (reference FS10808, diluted in 6.25% DMSO in PBS). Cyclophosphamide was purchased from Sigma-Aldrich^®^ (reference C0768, diluted in saline).

In ovo TCAM assay: After randomization at ED10 based on initial egg weight at ED0, treatments were administered twice onto the tumors or the CAM at days ED10 and ED13. Dosing was extrapolated on the basis of the average weight of all eggs. The range of doses used in ovo was estimated on the basis of those used in vivo. Treatments were applied topically directly onto the silicone ring-containing tumors using a volume of around 100 µL by pipetting. Cisplatin was used at 0.2, 0.4, 1, 2, 4, or 10 mg/kg. Sorafenib was used at 2 mg/kg. Doxorubicin was used at 0.4 mg/kg. Cyclophosphamide was used at 1, 10, or 100 mg/kg. TMZ was used at 1 mg/kg. 5-FU was used at 1 mg/kg.

In vivo subcutaneous models: Once the tumors reached an approximate volume of 100 mm^3^, the mice were randomized on the basis of their tumor volume. Mice were treated with cisplatin at 1 mg/kg five times a week or at 3 mg/kg three times a week via the intraperitoneal route (i.p.) [23,24], with cyclophosphamide at 100 mg/kg once a week via the i.p. route, or with temozolomide at 10 mg/kg five times a week via the oral route [25,26], both starting from the randomization and until the end of the experiment.

### 2.7. Statistics

The procedure was adapted from [21,22]. Statistical analysis and graphical representations were performed using GraphPad Prism (version 9.2.0). A *p*-value <0.05 was considered statistically significant (* *p* < 0.05; ** *p* < 0.01; *** *p* < 0.001; **** *p* < 0.0001). All data per group were checked for normality using the D’Agostino–Pearson test. In cases of a nonsignificant difference, a parametric test was used; in cases of a significant difference, a nonparametric test was performed. For the tumor CAM assay, data were analyzed using a Student *t*-test or Mann–Whitney test. For tumor mouse models, tumor volume was analyzed using a two-way ANOVA or mixed models (groups and days as factors) with repeated measures on each day. In the case of a significant group and/or interaction effect, post hoc Bonferroni’s multiple comparison tests (for each day) were performed. The cumulative survival distribution was constructed using the Kaplan–Meier method. Differences between survival curves were tested for significance with the log rank test.

## 3. Results

### 3.1. Comparison of Toxicity of Anticancer Drugs in Rodent Tumor Graft and TCAM Model

We compared the safety profile of chemotherapeutic drug agents between mouse and chicken embryos. We first analyzed the response of a single dose administered at ED7 in naïve chicken embryo. Acute treatment of 10 mg/kg also rapidly induced the overall lethality of embryos after only 2 days of monitoring. The acute treatment of 2 mg/kg led to partial lethality, while the dose of 0.2 mg/kg was not lethal (Figure 2A). While such analysis provides a response regarding the toxicity of a compound on chicken embryo, the use of naïve embryos did not reflect the inflammatory situation when tumor occurred. Indeed, tumoral processes such as drug metabolism have to be considered in order to assess a potential toxicity due to intratumoral activation of prodrugs or other activity modulation [27,28]. The use of physiopathological-based models such as a xenograft model in order to consider the tumor context could improve predictive potential toxicological in a relevant context [29]. Thus, we investigated the effect of different doses of cisplatin in CT26 tumor-bearing chicken embryos after repeated administration (Figure 2B). We demonstrated that the administration at ED10 and ED13 with a dose of 10 mg/kg induced a complete lethality of embryos after chronic administration of cisplatin. We also observed lethality at 4 mg/kg with 20% survival and at 1 mg/kg with 50% survival (Figure 2B). Conversely, doses of 0.2 and 0.4 mg/kg did not affect embryo survival (Figure 2B). The analysis of repeated treatment in mice demonstrated that mice started to significantly lose body weight from day 21 with the dose of 2 mg/kg and from day 29 with the dose of 1 mg/kg, indicating some toxicity (Figure 2C). The survival analysis also showed that only the dose of 2 mg/kg induced lethality (20%; Figure 2D). We also evaluated cyclophosphamide toxicity in both species. As previously done for cisplatin, we first evaluated cyclophosphamide as single administration in naïve embryos. In this context, the dose of 100 mg/kg of cyclophosphamide exerted a strong lethality in chicken embryos starting from day 3 with no survival. Comparatively, doses of 1 and 10 mg/kg did not affect survival (Figure 2E). Moreover, we observed similar toxicity when cyclophosphamide was administered twice in tumor-bearing chicken embryos (Figure 2F). Comparatively, although no lethality was observed in mice treated with 100 mg/kg of cyclophosphamide (Figure 2G), we observed a reduction in body weight in treated mice compared to vehicle mice (Figure 2H). No toxicity was observed at 10 mg/kg (data not shown) as was observed in chicken embryos. Globally, chicken embryos demonstrated higher lethality as compared to mice at similar doses. Importantly, this statement might be nuanced taking into account the embryo weight growth during embryonic development from an embryo weight relative to egg weight that passed from around 2–3% at ED9 to more than 70% at ED21 [30]. In this case, the dosing used here is at least 10-fold more important in embryo when compared to full egg weight. This might also explain why chicken embryos displayed higher lethality at the same dose used in mice. Thus, the TCAM model may serve as a guide for dosing of test substances, while avoiding the unnecessary toxicity for mouse xenograft models.

### 3.2. Tumor Response to Anticancer Drugs in TCAM Assay

We generated tumors from different types using glioblastoma, melanoma, triple-negative breast cancer, prostate cancer, colorectal cancer, liver cancer, and lung cancer cell lines (Appendix A). We observed different types of tumor shape as a function of the cell line used, from a round (Appendix A) or flatter shape (Appendix A) to smaller structures (Appendix A). Thanks to the tumor generated, we demonstrated that these multiple tumor types could be evaluated with different drugs, represented as a heatmap (Figure 3). The doses selected were based on the maximal nonlethal dose, and the type of treatment per cell type was based on clinical practice and new indications described in the literature. We showed that temozolomide (TMZ), the standard of care (SOC) used in glioblastoma (GBM) and in melanoma [31,32], significantly reduced the weight of GL261 mouse GBM tumors (Figure 3 and Appendix A), U118MG human GBM tumors (Figure 3 and Appendix A), and A375 human melanoma tumors (Figure 3 and Appendix A). We also challenged the effect of cisplatin, which is a chemotherapy used in different tumor indications [33,34]. We demonstrated that cisplatin significantly reduced 4T1 mouse triple-negative breast cancer (TNBC) and MDA-MB-231 TNBC tumor weight (Figure 3 and Appendix A). Cisplatin also significantly reduced the weight of LNCaP human prostate tumors (Figure 3 and Appendix A), of HCT-116 and HCT-8 human colorectal tumors (Figure 3 and Appendix A), of CT26 mouse colon tumors (Figure 3 and Appendix A), of A375 human melanoma tumors (Figure 3 and Appendix A), and of Hep G2 human liver tumors (Figure 3 and Appendix A). Conversely, cisplatin did not affect PC-9/CR human lung tumor weight (Figure 3 and Appendix A). We also evaluated the effect of doxorubicin which is an antibiotic agent that inhibits DNA topoisomerase II and induces DNA damage [35,36]. Doxorubicin is used against a wide range of cancers such as carcinomas, sarcomas, and hematological cancers [37]. We showed that doxorubicin significantly reduced the weight of 4T1 mouse TNBC tumors (Figure 3 and Appendix A), of PC-3 human prostate tumors (Figure 3 and Appendix A), of HCT-116 and HCT-8 human colorectal tumors (Figure 3 and Appendix A), of CT26 mouse colon tumors (Figure 3 and Appendix A), and of Hep G2 human liver tumors (Figure 3 and Appendix A). 5-Fluorouracil (5-FU) inhibits DNA/RNA synthesis and is widely used in clinic in several tumor indications including the colon [38]. We demonstrated that 5-FU did not affect MDA-MB-231 human TNBC tumor weight (Figure 3 and Appendix A), but reduced HCT-8 human colorectal tumor weight (Figure 3 and Appendix A). Lastly, we analyzed the effect of sorafenib, which is a kinase inhibitor targeting different signaling pathways such as Raf or VEGF receptors. Sorafenib affects different cancer-associated events such as apoptosis and angiogenesis [39,40]. We demonstrated that sorafenib did not affect GL261 mouse GBM or A375 melanoma tumor weight (Figure 3 and Appendix A), while it had a nonsignificant minor effect on MDA-MB-231 human TNBC and HCT-8 human colorectal tumor weight (Figure 3 and Appendix A). Furthermore, we showed that cisplatin reduced PC-9 human lung tumor weight oppositely to PC-9/CR (Figure 2 and Appendix A). It is also interesting to note that embryo death was limited in the control groups treated with vehicle with less than 10% of death, even if some disparity existed between the tumor models.

### 3.3. Comparison of Tumor Response to Anticancer Drugs in Rodent Tumor Graft and TCAM Models

We demonstrated that multiple tumor types could be evaluated with different drugs in the TCAM model (Figure 3). We then directly compared the response of chemotherapeutic drug agents between mouse and chicken embryos in order to evaluate the relevance of the TCAM model as compared to the gold-standard efficacy xenograft rodent models. We first demonstrated, as similarly reported above (Appendix A), that temozolomide significantly reduced the weight of U118MG human GBM tumors in ovo (Figure 4A), as well as significantly repressed tumor volume in vivo in mice (Figure 4B). Similar analysis was performed using cisplatin, which significantly reduced the weight of CT26 mouse colon tumors in ovo (Figure 4C). Cisplatin also significantly repressed tumor volume in vivo in CT26-bearing mice (Figure 4D). Moreover, cisplatin significantly reduced the weight of 4T1 mouse TNBC tumors in ovo (Figure 4E). Cisplatin also significantly repressed tumor volume in vivo in 4T1-bearing mice (Figure 4F). Conversely, cisplatin did not affect cisplatin-resistant PC-9/CR human lung cancer in ovo (Figure 4G and Appendix A) or in vivo in mice (Figure 4H). No mouse deaths occurred during the in vivo xenograft experiment (Figure 4).

## 4. Discussion

In this study, we demonstrated that the TCAM model can be used for early screening in a large cohort of tumor types to evaluate multiple anticancer agents and may be a substitute to the gold-standard mouse subcutaneous xenograft model. Both efficacy and safety concerns can be investigated with the corresponding drug response. To the best of our knowledge, there have been very few direct comparisons of tumor progression in TCAM and murine models [41,42,43,44,45]. Indeed, Strojnik and collaborators compared the architecture of glioblastoma tumors in chicken embryos and mouse models [41]. Lyu and collaborators compared the expression of u-PAR and tumor growth profile between chicken embryos and mouse models [43]. Aguirre-Ghiso and collaborators compared p38/ERK activation in carcinoma cells [42]. In these three articles, no pharmacological evaluation comparison was performed. Conversely, Weiss and collaborators showed some similarity of response of A2780 human ovarian tumors upon combination of antiangiogenic therapies in chicken embryos and in mouse xenograft models, and they also showed that RAPTA-C, a ruthenium-based compound, is able to reduce tumor growth in both chicken embryo and in mouse xenograft models at a respective dose of 0.2 mg/kg (dose extrapolated on the basis of embryo weight) and 100 mg/kg [44,45]. Outside of these two articles from Weiss and collaborators, we did not find additional articles comparing drug response in mouse and chicken embryo models. Thus, the present face-to-face evaluation of anticancer drugs presents original results with sufficient scaling in terms of the number of cancer cell lines and drugs screened to demonstrate the reproducibility and reliability of the TCAM model as an alternative to mice.

In our work, we tested in our experiments two drugs in both mouse xenograft and TCAM models, TMZ which is the SOC for GBM [32] and cisplatin, a chemotherapy extensively used for treating lung, testicular, prostate, or ovarian cancer patients [33,46]. Moreover, platinum-based chemotherapies for TNBC, liver cancer, melanoma, or colorectal cancer demonstrated renewed and potent interest at the preclinical [21,33,47,48] and clinical level [49,50,51,52,53]. Since clinical evidence demonstrated tumor cell resistance to cisplatin treatment in patients, several sensitization strategies that might circumvent drug resistance have to be tested at a preclinical level and have to be validated in vivo [51]. Thus, the TCAM model represents an avenue for drug screening by proposing sensitive and resistance models of cancer. Our data demonstrate a similar anticancer response to cisplatin of 4T1 and CT26 breast and colon tumor both in chicken embryos and in mouse models (Figure 3). Interestingly, we observed that both sensitivity and resistance to cisplatin could be observed in both models. Indeed, we did not observe a therapeutic effect of cisplatin in the resistant PC-9 cell line lung tumor model xenograft in TCAM or in mouse (Figure 3). Moreover, we demonstrated, through the U118MG GBM cell line described to be poorly responsive to TMZ in the mouse xenograft model [54], that TMZ induced tumor reduction in both mouse and TCAM models with a similar tumor growth inhibition pattern (Figure 3). Thus, the TCAM might represent an interesting alternative to murine models in order to decipher drug resistance [18].

Chicken embryos could also be a relevant model for nonclinical safety pharmacology for acute or sub-chronic treatments [6,13,55]. Our results using cisplatin and cyclophosphamide in ovo partially recapitulated some observations in vivo, and the fate of chicken embryos predicted observable drug toxicity in mice, usually observed as body weight reduction over the treatment period. Safety analysis in chicken embryos may, therefore, serve as a basis to identify toxic doses of compounds prior to rodent experiments. The first dose of a treatment affecting embryo survival, for example, may be used as the highest dose administered in mice in order to limit unnecessary physiological stress. The present face-to-face comparison indicates a reasonable similarity in drug effects in the TCAM and mouse xenograft models. Altogether, these data suggest that the TCAM model could serve to test new therapeutic approaches in ovo with a similar response in terms of anticancer efficacy and of toxicity as compared with murine models.

One constraint of the TCAM model is the importance of embryonic death after manipulation of the egg, as described in the literature [56,57,58]. Several key steps in our protocol most likely account for the improved embryonic survival, as well as increased reliability of tumor growth. Indeed, with our protocol, embryo death was limited in the control groups treated with vehicle with less than 10% of death. Death observed might be imputable to egg manipulation or as a result of tumor burden. Moreover, we observed almost 100% tumor burden with the cellular models used. These data are in correlation with our data in mice, where all the cellular models used in ovo also generated tumors after subcutaneous engraftment in mice (data not shown). Thanks to our preliminary studies (data not shown), we identified several critical steps that may influence the effectiveness of the model including (i) the method for dropping the CAM away from the shell by applying depression through the air sac, (ii) the use of ethanol solutions to clean the shell to limit contamination, (iii) the use of Matrigel™ as a pro-survival scaffold for tumor cell growth, (iv) the use of a sterile silicone ring to concentrate the cells and permit rapid tumor growth, (v) the use of a stable concentration of two million cells, and (vi) the limited handling of the eggs during the procedure.

Alternative measurements beyond tumor weight can also be relevant to assess tumor progression, including tumor size, tumor cell count, tumor angiogenesis, or lung and liver metastasis within the embryo [1,5,6,17,18]. Additional analysis using in vivo imaging methods, with reporter cells or immunohistochemistry, allow for better characterization of tumor progression [3,59]. The CAM xenograft model represents a cost-effective way of quickly obtaining data in a setting that is more biologically relevant than cells grown in culture, and it is a valuable intermediate step in bridging pure in vitro work with more complex models of cancer, such as orthotopic animal models.

The evaluation of tumor volume and administration of treatments in rodents induces discomfort and stress due to handling, restriction, intravenous/intraperitoneal inoculation, or sedation [60]. In contrast, the CAM is not considered as challenging from an ethical point of view, because the chicken embryos do not react to nociceptive stimulations during most of their period of development. In fact, analysis of pain due to electric shock demonstrated an absence of reactions in chicken embryos until ED15 [61]. Moreover, chicken embryos do not show a sustained electroencephalography activity associated with pain perception before ED17 [8]. This absence of functional nociception during the first stages of development renders TCAM experimentation ethically more acceptable because it limits the influence of humane endpoints such as pain and stress on the experimental outcomes. This procedure could, therefore, be considered as ethically preferable to standard mouse experiments in order to obtain first preclinical in vivo proof of concept.

One could wonder if the short therapeutic window can be considered as a disadvantage for TCAM experiments, since the hatching of the chicken appears on day 21, and the termination of the monitoring in this work occurred at ED17. In contrast, several rodent experiments usually permit longer therapeutic windows. Nevertheless, this is not the case for all the tumor cell lines. Indeed, our TCAM model allows a monitoring of 10 days for tumor progression. One mouse model, PC-9/CR xenograft, did allow at least 30 days therapeutic window due to relatively low growth profile (Figure 4H), but other models such as U118MG xenograft and CT26 syngeneic models offered only 7 and 13 days, respectively, before animal sacrifice for ethical consideration (Figure 4B,D). Therefore, the 10 day period of monitoring in ovo seems comparable to several mouse models depending on the tumor cell lines.

Another limitation may be that the metabolism of drugs might be different between mouse and TCAM models. Indeed, the administration routes between mice and chicken embryos are different with intravenous/i.p./per os administration in mice versus topical administration on CAM for chicken embryos in our protocol. Nevertheless, xenobiotic metabolism has been described in embryos following topical administration. Temozolomide, for example, administered into the circulation undergoes rapid nonenzymatic conversion at physiologic pH to the active compound monomethyl 5-triazino imidazole carboxamide (MTIC) once in circulation [62] and induces tumor growth repression in the TCAM model. In the case of complex and organ-specific enzymatic metabolism, a correct evaluation of the compound metabolic activation should be carefully investigated, and an alternative route of administration other than topical could be used. Intravenous or intra-allantois injection of compound could represent a correct and feasible alternative [63,64].

An important aspect using animal models is to appropriately select the dose range to translate the data between species. Indeed, for human, doses are often expressed as the quantity per body surface area (mg/m^2^). For instance, TMZ, the standard of care for GBM, is administered from 75 to 200 mg/m^2^ to patients [32], which corresponds to roughly 2 to 6 mg/kg [65,66]. Cisplatin is used at around 40 to 300 mg/m^2^ for the treatment of lung cancer in patients [46,67], which corresponds to roughly 1 to 8 mg/kg [65,66]. In this study, TMZ was administered onto the tumors at 1 mg/kg, while cisplatin was administered at 0.4 mg/kg. Considering the average embryo weight at ED10 to ED13, around 7 g [30], the actual dose given to the embryo should be around 0.1–0.2 mg/kg. Nevertheless, conversion between species based on mg/m^2^ cannot be directly applied for drugs administered by topical administration as is the case for TCAM model [65]. Thus, it is not possible to precisely extrapolate the correspondence of human doses, even if the range of dosing based on whole egg weight is quite similar. The doses inducing a pharmacological response in TCAM were globally 2–10-fold lower compared to mice. Indeed, in mice, TMZ was administered at 10 mg/kg orally and cisplatin was administered at 1 mg/kg or 2 mg/kg intraperitoneally, consistently with the doses used in human without correction.

Altogether, the TCAM model can, therefore, serve as a replacement method for rodent experiments, meeting the ethical obligations and 3Rs guideline for animal usage in tumor research, while still having a predictive response to anticancer drugs. The TCAM assay remains a versatile model for drug discovery, including screening, efficacy, and toxicity approaches [5,13]. A large variety of cancer types can be investigated in the TCAM xenograft model. Tumor growth and compound efficacy can be efficiently assessed by tumor weight measurements. Additional parameters, including targets or mechanisms of action, can be assessed by analysis of tumor markers and molecular pathways [18]. Tumor angiogenesis can also be evaluated by in vivo imaging [11]. The TCAM assay is a cost- and time-effective method for producing predictive data for anticancer drug development and is recommended as a transitional in vivo tool for early screening of anticancer treatments, bridging in vitro tumor cell cultures to more complex procedures, such as orthotopic rodent models.

## 5. Conclusions

The TCAM assay is a cost- and time-effective method for producing predictive data for anticancer drug development and is recommended as a transitional in vivo tool for early screening of anticancer treatments, bridging in vitro tumor cell cultures to more complex procedures, such as orthotopic rodent models.

## Figures and Tables

**Figure 1 cancers-14-03548-f001:**
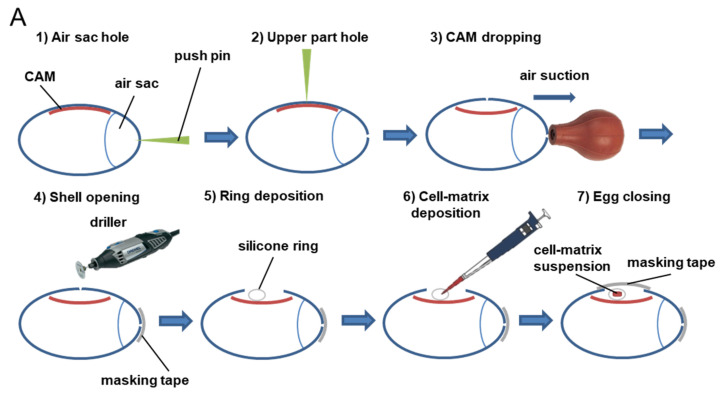
Procedure for evaluation of drug response and toxicity in tumor CAM model. (**A**) Using a sterile push pin, one hole was made through the air sac of eggs (1), and another one was made on the upper part of the eggs (2). The CAM was then dropped away from the shell by applying suction against the air sac hole using a safety bulb (3). An opening was made using a driller (4). A silicon ring was dropped onto the CAM (5), and a cell–matrix suspension was pipetted into the center of the ring (6). The eggs were retaped and moved back to the incubator (7). (**B**) Chicken embryos were incubated at 37.7 °C and 60% humidity upon arrival and until day 7 post-fertilization. At ED7, a window on each egg shield was created, and tumor cells were deposited onto the CAM. The eggs were incubated for 10 additional days, and, during this period, developing tumors were treated with test substances at days ED10 and ED13. At ED17, viable embryos were sacrificed, and tumors were collected and weighted to analyze the potential antitumor effects of test substances. The toxicity of anticancer treatments was evaluated by the percentage of embryos that died during the therapeutic window.

**Figure 2 cancers-14-03548-f002:**
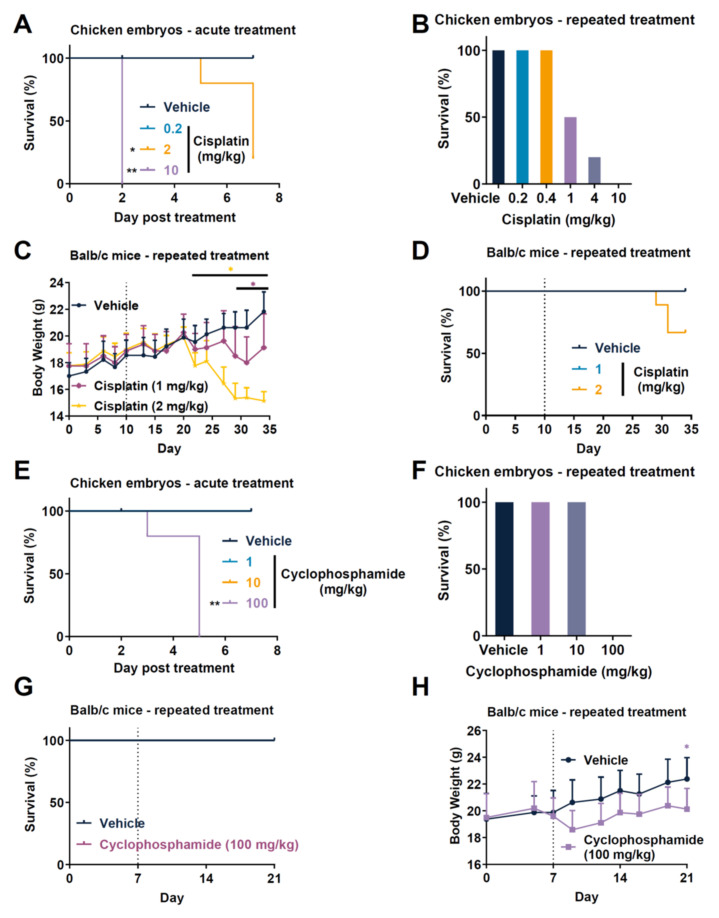
Comparative toxicity of cisplatin and cyclophosphamide in ovo and in vivo mouse models. (**A**–**D**) Analysis of toxicity induced by cisplatin in chicken embryos or in mice. Treatment with cisplatin was performed at day 7 post egg fertilization (defined as day 0) in nongrafted chicken embryos (naïve), and survival was analyzed during a 7 day period (**A**). Treatment with cisplatin was given at days 10 and 13 post egg fertilization to CT26 tumors-bearing chicken embryos. and survival was analyzed at day 17 (**B**). Treatment with cisplatin was given five times per week at 1 mg/kg or twice per week at 2 mg/kg from day 10 to Balb/c mice grafted with CT26 tumors, body weight was measured (**C**), and survival was analyzed (**D**). (**E**–**G**) Analysis of toxicity induced by cyclophosphamide in chicken embryos and in mice. Treatment with cyclophosphamide was performed at day 7 post egg fertilization (defined as day 0) in nongrafted chicken embryos (naïve), and survival was analyzed during a 7 day period (**E**). Treatment with cyclophosphamide was given at days 10 and 13 post egg fertilization to CT26 tumors-bearing chicken embryos, and survival was analyzed at day 17 (**F**). Treatment with cyclophosphamide was given three times a week at 100 mg/kg from day 7 to Balb/c mice grafted wit’h 4T1 tumors, survival was monitored (note that no mouse death occurred during the study) (**G**), and body weight was measured (**H**). Survival curves constructed using the Kaplan–Meier method and analyzed by log rank test (vs. control, * *p*  ≤  0.05, ** *p*  ≤  0.01). Body weight was compared by a two-way ANOVA test or mixed model followed by Bonferroni’s comparison test (* *p* ≤ 0.05); *n* = 5 embryos per group (**A**,**G**); *n* = 10 embryos per group (**B**,**F**); *n* = 9 mice per group (**C**,**D**); *n* = 8 mice per group (**G**,**H**). Data represent the mean and SD (**C**,**H**).

**Figure 3 cancers-14-03548-f003:**
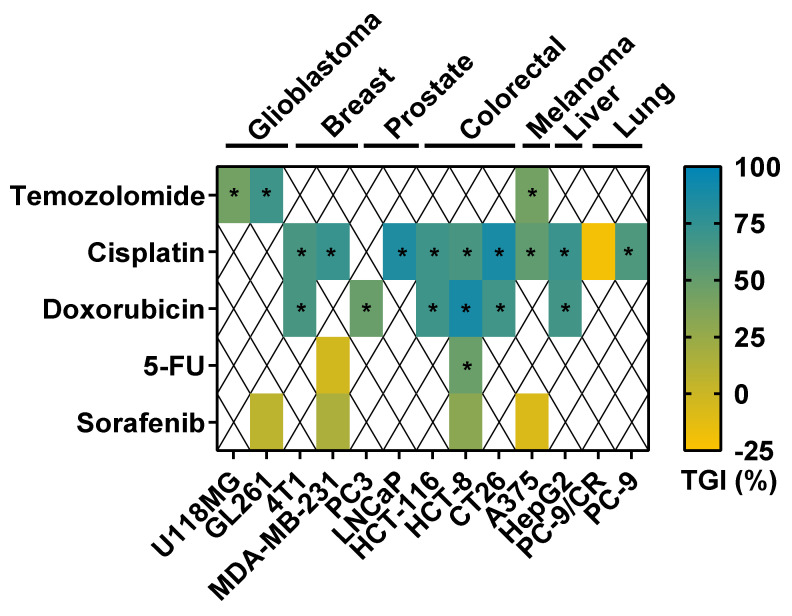
Tumor response to anticancer drugs in the TCAM model. Heatmap representation of tumor growth inhibition (TGI, as a percentage) per test substance and cell type, demonstrating variability of tumor cell responses upon anti-cancer drugs. Analysis was performed by measuring tumor weight after 10 days of growth. Blue represents high tumor inhibition, whereas yellow represents the absence of an anticancer effect. Two administrations (days 10 and 13) onto the tumors of cisplatin at 0.4 mg/kg, sorafenib at 2 mg/kg, doxorubicin at 0.4 mg/kg, temozolomide at 1 mg/kg, and 5-FU at 1 mg/kg were performed. A significant difference as compared to vehicle-treated group using Student *t*–test or Mann–Whitney test is represented by * (*p* < 0.05). See related data in Appendix A and Figure 4.

**Figure 4 cancers-14-03548-f004:**
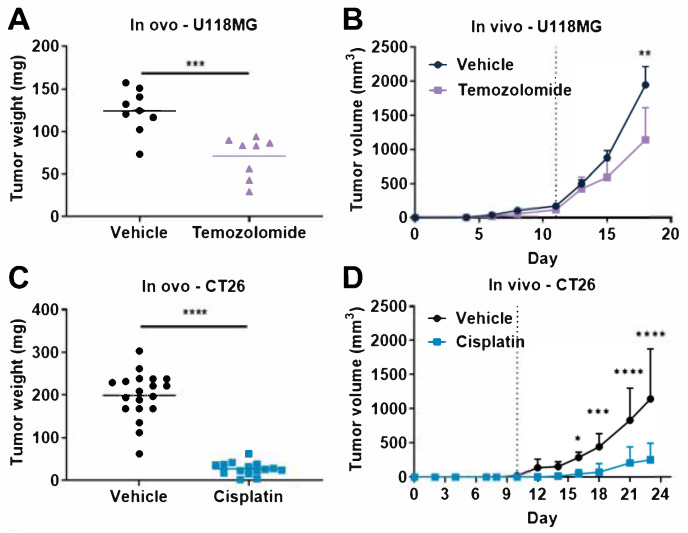
Face-to-face comparison of anticancer effect of cisplatin and temozolomide in ovo and in vivo xenograft models. (**A**,**B**) Temozolomide antitumor activity was evaluated on U118MG human glioblastoma tumors in ovo using the tumor CAM assay (**A**) and in subcutaneous xenograft model in Balb/c nude mice (**B**). (**C**,**D**) Cisplatin antitumor activity was evaluated on 4T1 mouse triple-negative breast tumors in ovo using the tumor CAM assay (**C**) and in subcutaneous syngeneic model in Balb/c mice (**D**). (**E**,**F**) Cisplatin antitumor activity was evaluated on CT26 mouse colorectal tumors in ovo using the tumor CAM assay (**E**) and in subcutaneous syngeneic model in Balb/c mice (**F**). (**G**,**H**) Cisplatin antitumor activity was evaluated on cisplatin-resistant PC-9 (PC9/CR) human lung tumors in ovo using the tumor CAM assay (**G**) and in subcutaneous xenograft model in Balb/c nude mice (**H**). In the tumor CAM assay, temozolomide (**A**) and cisplatin (**C**,**E**,**G**) were administered onto the tumors at 1 and 0.4 mg/kg, respectively. In vivo in mice, temozolomide was administered orally at 10 mg/kg five times a week (**B**) and cisplatin was administered intraperitoneally at 1 mg/kg five times a week (**D**,**H**) or 2 mg/kg three times a week (**F**). Note that the percentage of embryo spontaneous death in vehicle group for embryos engrafted with U118MG tumors was 0% (0/9) (**A**), for embryos engrafted with CT26 tumors was 5% (1/20) (**C**), for embryos engrafted with U118MG tumors was 0% (0/15) (**E**), and for embryos engrafted with PC9/CR tumors was 17% (1/6) (**G**). A discontinuous line highlights the treatment beginning. Student *t*-test, Mann–Whitney (in ovo), or two-way ANOVA test followed by Bonferroni’s comparisons test (in vivo) was performed (* *p*  ≤  0.05, ** *p*  ≤  0.01, *** *p*  ≤  0.001, **** *p* ≤  0.0001). Each egg is represented by a symbol in the in ovo graphs; *n* = 6 (**B**), *n* = 10–12 (**D**), *n* = 7–8 (**F**), and *n* = 5–6 (**G**) mice per group. Data represent the mean or mean and SD.

## Data Availability

The data presented in this study are available on reasonable request from the corresponding author.

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
