# Peer review of "A Face-To-Face Comparison of Tumor Chicken Chorioallantoic Membrane (TCAM) In Ovo with Murine Models for Early Evaluation of Cancer Therapy and Early Drug Toxicity"

_cancers, 2022, doi:10.3390/cancers14143548_

Round 1

Reviewer 1 Report

1. This manuscript compares tumor chicken chorioallantoic membrane (TCAM) in ovo model and murine model for early evaluation of cancer therapy and drug toxicity. The authors screened 5 anti-cancer agents in 12 tumor cell lines and four human and mouse tumor models. 

2. The TCAM model has high potential to replace rodent experiments in tumor research, drug discovery or toxicity study. The results of present study shows interesting points. However, there are some points need to be further clarified.

3. In page 20, line 608, Fingolimod seems not related to present study. Please check again if this is correct or not.

4. In page 20, authors need to re-arrange the orders of figure 2 A to 2F. Present sequences of Figure 2 are hard to show the toxic effects of cisplatin and cyclophosphamide on chicken embryo and Balb/c mice.

5. In page 10 and page 20, it is suggested to provide:

5.1 the survival of cyclophosphamide repeated treatment on chick embryo

5.2 the survival of cyclophosphamide repeated treatment on Balb/c mice

6. Please provide more discussion on the dose conversion among chicken embryo, mice model, and human doses.

Author Response

  1. This manuscript compares tumor chicken chorioallantoic membrane (TCAM) in ovo model and murine model for early evaluation of cancer therapy and drug toxicity. The authors screened 5 anti-cancer agents in 12 tumor cell lines and four human and mouse tumor models. 
  2. The TCAM model has high potential to replace rodent experiments in tumor research, drug discovery or toxicity study. The results of present study shows interesting points. However, there are some points need to be further clarified.

Author’s response: We gratefully thank the reviewer for recognizing our work and the different advices and comments that will improve our manuscript. Thanks to the reviewer’s comments, we improved the manuscript and we hope that the new version provides additional and requested information.

  1. In page 20, line 608, Fingolimod seems not related to present study. Please check again if this is correct or not.

Author’s response: We thank the reviewer to notice this mistake, indeed Fingolimod is irrelevant since we evaluated the effect of Cyclophosphamide. The text is revised accordingly (line 302).

  1. In page 20, authors need to re-arrange the orders of figure 2 A to 2F. Present sequences of Figure 2 are hard to show the toxic effects of cisplatin and cyclophosphamide on chicken embryo and Balb/c mice.

Author’s response: We are agreeing with the reviewer that the presentation of the results and the dose schedules was a bit confusing. We edited the figure panels and the text in order to clarify this. We explicitly mentioned the difference between single done in naïve non grafted chicken embryos and repeated (2) administration done in a tumor context in tumor-bearing embryos. We also include complementary data with repeated treatment of Cyclophosphamide in 4T1 tumor-bearing chicken embryos in order to be consistent with Cisplatin (see below). The text and figure are revised accordingly (lines 262-274 and 302 to 320 and Figure 2).

  1. In page 10 and page 20, it is suggested to provide:

5.1 the survival of cyclophosphamide repeated treatment on chick embryo

Author’s response:  We are agreeing with the reviewer that these data were missing in the previous version of our manuscript. We are now pleased to include new data generated in which 4T1 tumor-bearing embryos were treated at ED10 and 13 and survival was monitored. These data demonstrated in a similar manner as the single administration in naïve embryos that Cyclophosphamide at 100 mg/kg I toxic whereas at 10 and 1 mg/kg not (lines 286 to 288). These data are currently presented in the new Figure 2, panel 2F.

5.2 the survival of cyclophosphamide repeated treatment on Balb/c mice

Author’s response: in the original version of our manuscript, no mouse death occurred during the study upon Cyclophosphamide treatment, but the data were not shown. We are agreeing with the reviewer that this data should finally be shown and is now included as Figure 2G.

  1. Please provide more discussion on the dose conversion among chicken embryo, mice model, and human doses.

Author’s response: We are agreeing with the reviewer that such information might be relevant in the scope of this study and we added a paragraph emphasizing the relationship between doses through species and the interest of TCAM in this approach (lines 526 to 541).

Reviewer 2 Report

In this work, the authors develop tumor xenograft models for a variety of cancer cell lines in the CAM model and use these models to screen a selection of anticancer agents in direct comparison to more traditional mouse xenograft models. The authors demonstrate reproducible tumor growth inhibition patterns in the TCAM models when compared to mouse xenograft models, demonstrating TCAM to be a viable alternative or complement to mouse models. The authors have clearly performed meticulous efforts categorize these tumor models and screen drug responses, including the responses of drug sensitive and drug resistant cancer types.

I strongly agree with the author’s assertion that TCAM represent an important preclinical model to bridge the gap between in vitro and in vivo assays and support the use of TCAM as an alternative to mouse xenograft models, respecting the 3R principals. As such, I think this work is important to raise awareness and use of this model. In the discussion, the authors state that they: “did not find additional articles comparing drug response in mouse and chicken embryo models” and that “face-to-face evaluation of anti-cancer drugs has not previously been published and presents original results”. This is not entirely true, as some direct comparisons have been published (see references in point 3 below). Nonetheless, the scale of the work performed here, in terms of the number of cancer cell lines and drugs screened, is still important in demonstrating the reproducibility and reliability of the TCAM model as an alternative to mice.  

Major points:

1)      Lines 229-230: Please provide a better definition and reasoning for selecting the acute and chronic treatment schedules in the CAM model. Chronic treatment amounts to 2 treatments (on ED 10 and 13) vs acute treatment appears to be a single treatment on ED 7? This is a bit confusing, as I understand that the tumor cells are also inoculated on ED7. Was the acute treatment screened in embryos without tumor inoculation in order to screen toxicity effect? The text seems to indicate that tumors were present in both cases, so this is unclear.  

2)      Line 299: An additional reference you may want to include in the discussion section include a review on the use of the CAM model for angiogenesis and cancer drug screening: Nowak-Sliwinska, Angiogenesis, 2014, The chicken chorioallantoic membrane model in biology, medicine and bioengineering.

3)      Line 303: There are a few additional references you missed that directly compare the tumor growth inhibition effects of different compounds in TCAM and mouse xenograph models:

1)      Weiss et al, Royal Society of Chem, 2014, In vivo anti-tumor activity of the organometallic ruthenium(II)-arene complex [Ru(η6-p-cymene)Cl2(pta)] (RAPTA-C) in human ovarian and colorectal carcinomas

2)      Weiss et al. Angiogenesis, 2015, Rapid optimization of drug combinations for the optimal angiostatic treatment of cancer

4)      In the discussion, the authors state: “Globally, chicken embryos demonstrated stronger toxicological response compared to mice with lower lethal doses.” I am not convinced this state is correct. This may be an artifact based on how the doses in each model were selected. The authors state that: “Dosing was extrapolated based on the average weight of all eggs.” It may be more relevant to consider the embryo’s estimated weigh on the treatment days (not the weight of the entire egg, this information can be found in literature) in order to define the dose administered.

5)      In the discussion, the authors state that they observed minimal embryo death in vehicle treatment groups and almost 100% tumor burden. Having previously worked with the CAM model, I find this very surprising. The CAM model is very sensitive to manipulations and even susceptible to death as a result of tumor burden. I think it would be prudent to report the % of embryo death in all groups. This information is important for this publication to serve as a reference for future studies exploring these models.

6)      In the same respect as the previous point, I think it is also important to report the rates of successful tumor inoculation for each cell line model and if any criteria were employed to consider if a tumor was successfully inoculated in the CAM model (i.e. was there a minimum tumor size selected before randomizing into groups, etc.). This information is reported for the mouse model but not TCAM. Agitation of the CAM membrane due to manipulations and placing the plastic ring on the surface can cause some thickening of the membrane and angiogenesis. Some of these cases may look like tumor growth in the first few days of treatment. Were these types of problems encountered and how were they assessed/ eliminated. In my experience, not all tumor cell lines grow on the CAM and each cell line has variable success during inoculation. I think this is an important point to discuss, especially if other cell lines were screened and did not successfully implant, these should be reported.

Minor points:

1)      Please explicitly state the route of drug administered in the TCAM model as being topical in the methods section. This only became clear to me when reading the discussion section. Also describe the volume of solutions administered and solvent used to dissolve them?

2)      Line 375: “for example, ‘administered’ …”

3)      Line 380: ‘of’ should be ‘or’, I believe.

4)      Line 611: should refer to part (A) for figure, not (D)?

5)      Line 633: Caption of Figure 3 says to refer to Figure 3?

6)      Line 646: Reference to figure parts (A) and (C).

7)      Line 646: Reference to figure parts (B).

Author Response

In this work, the authors develop tumor xenograft models for a variety of cancer cell lines in the CAM model and use these models to screen a selection of anticancer agents in direct comparison to more traditional mouse xenograft models. The authors demonstrate reproducible tumor growth inhibition patterns in the TCAM models when compared to mouse xenograft models, demonstrating TCAM to be a viable alternative or complement to mouse models. The authors have clearly performed meticulous efforts categorize these tumor models and screen drug responses, including the responses of drug sensitive and drug resistant cancer types.

I strongly agree with the author’s assertion that TCAM represent an important preclinical model to bridge the gap between in vitro and in vivo assays and support the use of TCAM as an alternative to mouse xenograft models, respecting the 3R principals. As such, I think this work is important to raise awareness and use of this model. In the discussion, the authors state that they: “did not find additional articles comparing drug response in mouse and chicken embryo models” and that “face-to-face evaluation of anti-cancer drugs has not previously been published and presents original results”. This is not entirely true, as some direct comparisons have been published (see references in point 3 below). Nonetheless, the scale of the work performed here, in terms of the number of cancer cell lines and drugs screened, is still important in demonstrating the reproducibility and reliability of the TCAM model as an alternative to mice. 

Author’s response: We gratefully thank the reviewer for recognizing the quality of our work and the different advices and comments that will improve our manuscript. Thanks to the reviewer’s comments, we improved the manuscript and we hope that the new version provides additional and requested information.

Major points:

  • Lines 229-230: Please provide a better definition and reasoning for selecting the acute and chronic treatment schedules in the CAM model. Chronic treatment amounts to 2 treatments (on ED 10 and 13) vs acute treatment appears to be a single treatment on ED 7? This is a bit confusing, as I understand that the tumor cells are also inoculated on ED7. Was the acute treatment screened in embryos without tumor inoculation in order to screen toxicity effect? The text seems to indicate that tumors were present in both cases, so this is unclear.

Author’s response: We gratefully thank and we are agreeing with the reviewer that presentation of the dose schedules was a bit confusing. We edited the figure 2 and the text in order to clarify this. We explicitly mentioned the difference between single and repeated (2) administration. The single administration was done in naïve non grafted chicken embryos whereas repeated administration was done in a tumor context in tumor-bearing embryos justifying the difference in schedule. Indeed, single administrations were done at ED7 on viable embryos whereas repeated administrations were done 3 days after tumor inoculation at ED10 and ED13. We described this in our edited manuscript (lines 262-274). We also included complementary data with repeated treatment of Cyclophosphamide in tumor-bearing chicken embryos in order to be consistent with Cisplatin. Indeed, we also estimated that these data were missing in the original version of our manuscript. We are now pleased to include new data generated in which 4T1 tumor-bearing embryos were treated at ED10 and 13 and survival was monitored. These data demonstrated in a similar manner as the single administration in naïve embryos that Cyclophosphamide at 100 mg/kg is toxic whereas at 10 and 1 mg/kg not. These data are currently presented in the new Figure 2, panel 2F (lines 286 to 288 and Figure 2).

  • Line 299: An additional reference you may want to include in the discussion section include a review on the use of the CAM model for angiogenesis and cancer drug screening: Nowak-Sliwinska, Angiogenesis, 2014, The chicken chorioallantoic membrane model in biology, medicine and bioengineering.

Author’s response: We gratefully thank the reviewer for identifying this interesting review that is now cited in our edited manuscript (lines 66 and 455).

3)      Line 303: There are a few additional references you missed that directly compare the tumor growth inhibition effects of different compounds in TCAM and mouse xenograph models:

1)      Weiss et al, Royal Society of Chem, 2014, In vivo anti-tumor activity of the organometallic ruthenium(II)-arene complex [Ru(η6-p-cymene)Cl2(pta)] (RAPTA-C) in human ovarian and colorectal carcinomas

2)      Weiss et al. Angiogenesis, 2015, Rapid optimization of drug combinations for the optimal angiostatic treatment of cancer

Author’s response: We gratefully thank the reviewer for identifying these two papers that we missed on our search. These two references are now included and discussed in the edited manuscript (lines 423 to 430).

  • In the discussion, the authors state: “Globally, chicken embryos demonstrated stronger toxicological response compared to mice with lower lethal doses.” I am not convinced this state is correct. This may be an artifact based on how the doses in each model were selected. The authors state that: “Dosing was extrapolated based on the average weight of all eggs.” It may be more relevant to consider the embryo’s estimated weigh on the treatment days (not the weight of the entire egg, this information can be found in literature) in order to define the dose administered.

Author’s response: We agree with the reviewer that the sentence “Globally, chicken embryos demonstrated stronger toxicological response compared to mice with lower lethal doses” needs some revisions. Our initial goal was to mention that chicken embryos are quite sensitive to drug toxicity but we did not extrapolate the real dosing on embryo weight. Indeed, the dynamic of embryo weight through rough estimation complexity the manner to select appropriate dosing, that is why we used a manner to transfer from a model to another a corresponding dose of treatment. Thus, we selected the whole egg weight with its per se limitation. We still think that this way is quite useful since it guided us to select appropriate efficient doses in both chicken embryos and mice.  In order to integrate the comment of the reviewer, we added a paragraph describing this limitation. “Globally, chicken embryos demonstrated higher lethality as compared to mice at the same doses. Importantly, this response might be to put in correlation with the embryos weight that is growing during the embryonic development from an embryo weight relative to egg weight that passed from around 2-3% at ED9 to more than 70% at ED21 [27]. In this case, the dosing used here is at least 10-fold more important in embryo when compared to full egg weight. This might also explain why chicken embryos displayed higher lethality at same dose used in mice.” (line 292 to 297).

  • In the discussion, the authors state that they observed minimal embryo death in vehicle treatment groups and almost 100% tumor burden. Having previously worked with the CAM model, I find this very surprising. The CAM model is very sensitive to manipulations and even susceptible to death as a result of tumor burden. I think it would be prudent to report the % of embryo death in all groups. This information is important for this publication to serve as a reference for future studies exploring these models.

Author’s response: We understand the comment of the reviewer. We think that the use of Matrigel™ as pro-survival scaffold for tumor cell growth and a stable concentration of cells participate to explain our success in engraftment. Indeed, we observed that a lower number of tumor cells for grafting reduced tumor burden at least in some cell lines such as PC9/CR or HCT-116. Moreover, all the models used in this study already demonstrated previous growing capabilities in mice prior to engraftment in ovo that could also explain why all our models also grew in ovo. Nevertheless, the tumor weight or the shape between our models could be very different depending of cell line used. We think that is information is important for the readers and we include additional pictures of the tumors in non-treated condition for each cell line in the new Figure S1 (lines 322-326). To be precised regarding our additional work, we also tested additional cellular models not used for pharmacological evaluation with similar success of engraftment including U138MG, M059J, and U87MG glioblastoma cells or LOVO colon cancer cells (data not shown). So far, we did not face any issue regarding tumor growth in ovo yet. Conversely, we already had bad experience with few cell lines that did not reach to tumor formation into mouse in our hand. Maybe, these cell lines might also poorly generate tumors into CAM, we will investigate on this issue within our further experiences.

Regarding the minimal embryo death observed, we think that our method with the limited handling of the eggs during the procedure could participate to the high survival rate we are observing. Indeed, once eggs are engrafted with tumor cells no more direct manipulation of the egg is done and treatments are administrated by moving the eggs thanks to a plastic scaffold. Thus, we observed a death rate corresponding to less than 10 % of the embryo (lines 359 to 361).  We also included the number of death reported in the vehicle group per experiment in the figure legend figure 4 and S2 (lines 403 to 406 and 777 to 785).

  • In the same respect as the previous point, I think it is also important to report the rates of successful tumor inoculation for each cell line model and if any criteria were employed to consider if a tumor was successfully inoculated in the CAM model (i.e. was there a minimum tumor size selected before randomizing into groups, etc.). This information is reported for the mouse model but not TCAM. Agitation of the CAM membrane due to manipulations and placing the plastic ring on the surface can cause some thickening of the membrane and angiogenesis. Some of these cases may look like tumor growth in the first few days of treatment. Were these types of problems encountered and how were they assessed/ eliminated. In my experience, not all tumor cell lines grow on the CAM and each cell line has variable success during inoculation. I think this is an important point to discuss, especially if other cell lines were screened and did not successfully implant, these should be reported.

Author’s response: As mentioned above, all the cell lines used in ovo were also used in mice with engraftment success. Nevertheless, we completely understood the point raised by the reviewer and we attenuated our statement mentioning this limiting step (lines 472-474). Indeed, as the reviewer mentioned it is important to state that not all tumor model could grow in ovo, even if our selected ones did. Regarding our randomization process, we decided to randomize at ED10 based on initial weight at ED0. Moreover, all the embryos selected had to be alive to be included, as identifiable by CAM blood perfusion. All viable eggs were randomized, no selection based only on tumor size was done. We think that this strategy may improve the survival rate of embryos by limiting any direct egg manipulation. Nevertheless, all the embryos included in the analysis presented tumors at ED17, suggesting a limited impact of the randomization process used here. As mentioned by the reviewer, we could observe some thickness/tissue modification of the CAM and in particular neo-angiogenesis due to the addition of the silicone ring, observable 1 or 2 days after deposition. Nevertheless, at ED17 we did not face difficulties to dissociate the tumors tissue from the CAM during our micro-dissection. Even if for small tumors, micro-dissection may be assisted by microscopy system. In order to provide additional information regarding the shape of the tumors generated in ovo after dissection, we also added a new Figure S1 included representative images of non-treated tumors at ED17 post-sampling (line 322 to 326 and 760 to 763).

Minor points:

  • Please explicitly state the route of drug administered in the TCAM model as being topical in the methods section. This only became clear to me when reading the discussion section. Also describe the volume of solutions administered and solvent used to dissolve them?

Author’s response: We gratefully thank the reviewer for identifying this missing information. We edited the text accordingly and described that “Treatments were applied topically directly onto the silicone ring-containing tumors using a volume of around 100 µL by pipetting” (lines 235 to 236). The solvents used for both mouse and chicken embryo studies are described in the section “Treatments” (lines 224-231).

2)      Line 375: “for example, ‘administered’ …”

Author’s response: The text is revised accordingly (line 518).

3)      Line 380: ‘of’ should be ‘or’, I believe.

Author’s response: The text is revised accordingly (line 522).

4)      Line 611: should refer to part (A) for figure, not (D)?

Author’s response: We thank the reviewer to notice this mistake that has been revised accordingly (lines 302 to 320).

5)      Line 633: Caption of Figure 3 says to refer to Figure 3?

Author’s response: It should refer to figure 4. The text is revised accordingly (line 388).

6)      Line 646: Reference to figure parts (A) and (C).

Author’s response: The text is revised accordingly (line 400).

7)      Line 646: Reference to figure parts (B).

Author’s response: The text is revised accordingly (line 402).
